# Determinants of Human Papillomavirus Vaccine Acceptance among Caregivers in Nigeria: A Fogg Behavior Model-Based Approach

**DOI:** 10.3390/vaccines12010084

**Published:** 2024-01-13

**Authors:** Sohail Agha, Drew Bernard, Sarah Francis, Aslam Fareed, Ifeanyi Nsofor

**Affiliations:** 1Behavior Design Lab, Stanford University, Stanford, CA 94305, USA; 2Behavioral Insights Lab, Seattle, WA 98136, USA; drew@binsighslab.com (D.B.); sarah@binsightslab.com (S.F.); ifeanyi@binsighslab.com (I.N.); 3Indus Hospital, Karachi 75190, Pakistan; aslamfareed2@gmail.com

**Keywords:** HPV vaccination, vaccine hesitancy, motivation, ability, social factors, Fogg Behavioral Model, behavior change interventions

## Abstract

Human papillomavirus (HPV) vaccine uptake among adolescent girls is critical to reducing the burden of HPV-related cancers in Nigeria. This study assesses the factors influencing caregivers’ acceptance of HPV vaccination for their charges, using the Fogg Behavior Model (FBM) as a theoretical framework. We analyzed cross-sectional data from 1429 caregivers of girls aged 9–17 in six Nigerian states, using a survey instrument based on the FBM. Participants were recruited via Facebook and Instagram advertisements and interviewed through Facebook Messenger in August and September 2023. The study received ethical clearance from Nigeria’s National Health Research Ethics Committee. We applied bivariate and multivariate analyses to assess the relationships between the caregiver’s perception of how likely their adolescent girl was to get vaccinated in the next 12 months and motivation, ability, social factors (such as discussions with family and friends), injunctive norms, previous COVID-19 vaccination, and respondents’ sociodemographic characteristics. Adjusted odds ratios derived from logistic regression analyses revealed that caregivers’ motivation and ability, as well as social factors, were significantly associated with their perception that the adolescent girl in their care would get vaccinated within the next 12 months. Our findings suggest that behavioral interventions tailored to enhance motivation, ability, and social support among caregivers could significantly increase HPV vaccine uptake among adolescent girls in Nigeria.

## 1. Introduction

Cervical cancer ranks as the second most prevalent cause of cancer-related mortality among women 15–44 in Nigeria. High-risk HPV serotypes 16, 18, 31, 35, 51, and 52 are all prevalent serotypes in Nigeria with serotypes 16 and 18 responsible for 67% of Nigeria’s cervical cancer cases [1]. Cervical cancer accounted for 10,600 preventable deaths in Nigeria in 2019 [2].

The HPV vaccine was initially introduced in Nigeria in 2009 [3]. Yet, more than a decade later, research conducted among Nigerian undergraduate college students has shown that the knowledge of HPV as the causative agent behind cervical cancer and awareness of HPV vaccination as a preventive measure has remained low [3]. Research has also consistently shown low levels of uptake of HPV vaccination among secondary and college students in Nigeria [4,5].

Several studies reflect a paradoxical situation: low awareness of HPV and the HPV vaccine among caregivers, high willingness to vaccinate female children among caregivers who are aware of the HPV vaccine, and low levels of actual HPV vaccination among adolescent girls. One study revealed that, although 89% of educated mothers expressed readiness to vaccinate their female child, only 7% of their daughters had received the vaccine [6]. This was ostensibly due to a lack of knowledge of vaccination sites, indicating an information deficit [7], and prohibitive vaccine costs. In other words, the ability of caregivers to have their adolescent girl vaccinated appears to have been a major factor behind low HPV vaccine uptake. At the same time, a pilot study that made the vaccine available in rural Nigeria encountered community resistance based on misconceptions about why the vaccine was being introduced. In this instance, targeted advocacy efforts were needed to overcome community resistance [8], suggesting that the motivation of caregivers and the influence of their network members may be important factors in driving HPV vaccine adoption.

Low levels of involvement among the caregivers of adolescent girls in vaccination efforts and limited knowledge of the vaccine among caregivers have been identified as potential barriers to vaccine adoption. Studies have highlighted the potential role of parents in HPV vaccine adoption in Nigeria and the importance of educating parents about cervical cancer prevention and the role they might play in vaccination efforts [3]. In addition to low awareness, the cost and limited availability of the HPV vaccine have served as major barriers against vaccine adoption [3].

In a decisive public health response, in October of 2024, the Nigerian government initiated the integration of the human papillomavirus (HPV) vaccine into the national health system, with the aim of immunizing 7.7 million adolescent girls ages 9–14 [9]. Adolescent girls, in this age group, are now eligible to receive single-dose HPV vaccines, considered highly efficacious against the HPV types responsible for about 70% of cervical cancer cases. Starting with a five-day mass-vaccination campaign across 16 states and the Federal Capital Territory, this new initiative will subsequently be absorbed into the national routine immunization schedule [9]. The government’s public health response has been heralded by the World Health Organization as a “pivotal moment” in Nigeria’s campaign to eradicate cervical cancer.

Despite the government’s effort to remove one of the primary barriers to access by providing the HPV vaccine free of charge, the effectiveness of this initiative may hinge upon how vaccine information is presented to decision makers at the household and community levels—caregivers of adolescent girls and community members. For example, healthcare professionals have cautioned against emphasizing the vaccine’s role in preventing sexually transmitted infections to avoid provoking resistance from religious leaders in the community who might fear an increase in sexual activeness among vaccinated girls [10]. A focus on the vaccine’s general health benefits rather than its role in STD prevention has been proposed for vaccine promotion campaigns.

The urgency of meaningful communication, social mobilization, and community engagement in enhancing the vaccine’s acceptability among caregivers cannot be overstated [11]. This requires first understanding and then addressing caregiver concerns regarding the HPV vaccine.

There appears to be a substantial gap between awareness and acceptability of HPV vaccines among caregivers and the adoption of HPV vaccines by adolescent girls. The COVID-19 pandemic has had a negative impact on immunization programs worldwide [12], with COVID-19 vaccine misinformation having a negative effect on uptake of other vaccines. Moreover, the widespread perception that HPV vaccination will increase young women’s sexual activeness highlights the need for understanding caregivers’ perceptions of the HPV vaccine in order to develop a communication strategy that is sensitive to the local context and encourages caregivers to have their children vaccinated [13,14,15].

Nigerian caregivers play an important role in decisions regarding their children’s health. We designed this study to understand caregivers’ beliefs as they relate to the adoption of the HPV vaccine. These behavioral insights will be helpful in designing effective interventions to promote HPV vaccine adoption in Nigeria.

Previous research has shown that interventions designed using behavior models are much more likely to be successful than interventions that are not [16]. We developed a research instrument based on the Fogg Behavioral Model (FBM). The FBM’s increasing application in public health interventions [17,18] reflects its utility and growing acceptance amongst practitioners.

## 2. Methods

### 2.1. Study Design

We conducted a cross-sectional survey employing a multistage sampling technique to ensure the inclusion of diverse populations within Nigeria, a country with 36 states and the Federal Capital Territory (FCT) of Abuja. Nigeria is categorized into six geopolitical zones: North West, North East, North Central, South West, South East, and South Central. While our survey is not representative of the whole of Nigeria, we increased the heterogeneity of the sample by selecting one state from each geopolitical zone based on socio-economic, religious, and ethnic diversity. We recognize that a significant proportion of Nigerians may not be reached through a social media survey or may not be interested in responding to the type of advertisements that we posted on Facebook and Instagram to encourage respondents to volunteer their participation in the survey. The states selected were the following: Sokoto from the North West zone, Bauchi from the North East zone, Niger from the North Central zone, Lagos from the South West zone, Anambra from the South East zone, and Rivers from the South Central zone. We also wanted to ensure that we selected states that were going to be part of the government’s initial vaccine push as well as states where the vaccine would be introduced later. The states of Bauchi and Lagos are among the 16 states participating in the initial rollout of the HPV vaccination campaign in Nigeria.

### 2.2. Sampling Strategy

The sample was divided into 120 distinct strata based six states, five age groups, gender (male and female), and locality (urban or rural). The recruitment strategy involved running 120 separate advertisement sets tailored to these strata using Meta’s digital advertising platforms (Facebook and Instagram).

### 2.3. Participant Recruitment and Data Collection

This study built upon and took advantage of the tremendous growth in use of digital research tools that has occurred recently. Recruitment was conducted through targeted advertisements on Facebook and Instagram using Virtual Lab, an open-source tool [19]. The ads were configured to reach potential participants across the strata. Meta Messenger’s Chatbot service was employed for survey administration, with an incentive of NGN 400 (approximately USD 0.87) in mobile credit for survey completion. The advertisements reached 667,907 individuals, resulting in 14,992 link clicks. Of those, 3243 initiated the survey, and 1429 provided responses to nearly all the survey questions. The total survey advertising cost, i.e., the cost of the ads on Facebook and Instagram, was USD 2492.77.

Participants had the flexibility of completing the survey in one session or over multiple sessions. Based on the difference between the survey start time and the survey end time, the median time to finish the survey was 8.81 min. The data gathered from 1429 respondents were used for the analyses.

### 2.4. Outcome Measure

The primary outcome measured was the respondent’s perception that the female child/children in their care would receive the HPV vaccine within the next 12 months. This metric is particularly relevant in the context of the HPV vaccine introduction supported by Gavi, which commenced on 24 October 2023. Survey data collection was conducted in August and September 2023 and provides a true baseline against which the program progress may be measured. A follow-up survey is planned for March 2024 to determine the level of vaccine uptake following its introduction.

### 2.5. Sample Size Determination

One consideration in determining the sample size was to have a sufficient number of respondents from Bauchi and Lagos to compare changes reported by caregivers in these two states to caregivers in the other four states where the HPV vaccine would be introduced later.

No prior estimate was available for HPV vaccination among adolescent girls in Nigeria. The sample size was calculated conservatively, based on a 50% prevalence of HPV vaccination and a margin of error of 6 percentage points. We assumed a design effect of 1.5 for our sample size calculations. Our calculation indicated that a sample of 700 would be sufficient to ensure a margin of error for HPV vaccination within six percentage points. Thus, a total sample size of 1400 was determined based on having 700 respondents from Bauchi and Lagos and 700 from the other four states, where vaccine introduction was planned for a later date.

### 2.6. Selection and Operationalization of Variables

Our study employed the Fogg Behavior Model (FBM) to select the key variables for analysis. The FBM posits that behavior happens when three factors co-occur: motivation, ability, and prompt [20]. Our study captured data on motivation and ability because of their significant impact on vaccine uptake behaviors in Nigeria, as demonstrated in several recent studies [16].

To gauge motivation, we asked participants to respond to the statement, “Getting the girl who is in my care vaccinated against HPV is important to me”. Those strongly agreeing with this statement were classified as having a high level of motivation. For ability, participants were asked to respond to the statement, “Getting the girl vaccinated against HPV is difficult”, and respondents who strongly disagreed with this statement were classified as having high ability. Our approach is consistent with the methodology utilized in previous research measuring these two constructs [16] and the established correlation between health behaviors, including between vaccination and motivation and ability [16].

Awareness of cervical cancer is a crucial factor influencing the acceptance of the HPV vaccine among caregivers [15]. The survey assessed knowledge of HPV and vaccination through the following five questions: “Have you heard of the HPV vaccine?”; “What does the HPV vaccine protect against?”; “At what age should the HPV vaccine ideally be given?”; “Who should the HPV vaccine be given to”; and “To what extent do you agree with the statement HPV is a relatively uncommon STD?”. Statistical analysis (described later) showed that the first four questions were part of a scale. We created a summary HPV vaccine knowledge score, which summed up correct responses to the first four questions.

Despite high awareness levels, a disconnect between knowledge of HPV and knowledge of cervical cancer has been documented [7,15]. Underlining the extent of cervical cancer prevalence in Nigeria, qualitative research findings from one prior study revealed participants’ awareness of the disease, with at least one woman in every focus group discussion knowing someone who had had cervical cancer [13]. We asked survey respondents, “Have you ever known anyone who had cervical cancer?”.

In a systematic review, Kutz emphasized the role of family and community discussions in mitigating HPV-related stigma in sub-Saharan Africa [21]. We assessed this by asking respondents, “How likely are you to discuss HPV vaccination with family or friends?”. Such conversations are considered influential in shaping vaccination behaviors, as perceived positive norms around vaccination may emerge from these conversations and increase vaccine uptake. To measure injunctive norms, we sought caregiver agreement or disagreement with the statement “Most of my family and friends approve of girls getting vaccinated for HPV”, with strong agreement indicating a prevailing injunctive norm supportive of HPV vaccination.

Although hypothesized links between COVID-19 vaccine hesitancy, misinformation, and HPV vaccine hesitancy exist, scant research has addressed this relationship, even in high-income countries. In their review of the literature on this topic, Vraga et al. identified only seven studies that explored this connection [22]. These studies, from high-income countries such as the US, Japan, and the UK, found a positive association between COVID-19 vaccine hesitancy and HPV vaccine hesitancy. We included the question “Have you received a COVID-19 vaccination?” and reverse-coded it as a proxy indicator for COVID-19 vaccine hesitancy.

We measured caregiver’s age, gender, level of education, and residence in urban versus rural areas, and conceptualized these as control variables. We used information from the Meta platform on respondents’ state of residence to create a binary variable distinguishing residence in Northern Nigeria versus Southern Nigeria and conceptualized this as a control variable.

### 2.7. Data Analysis

Univariate analysis was conducted to provide frequency distributions on factors associated with HPV vaccine uptake, including motivation, ability, discussion of the HPV vaccination, family approval of HPV vaccination, and having received a COVID-19 vaccine. Frequency distributions were also examined on all control variables. Factor analysis was conducted to identify whether all HPV vaccine knowledge variables were part of the same underlying construct. Four out of five knowledge variables had a high factor loading on the first principal component, which explained 40% of the variance. We also conducted reliability analysis using these four HPV vaccine knowledge variables. The Cronbach’s Alpha statistic was 0.75, which is considered acceptable, and a summary score was created to measure HPV knowledge, as described earlier.

Bivariate analysis was conducted to explore the relationships between potential correlates of HPV vaccination and the caregiver’s perception that their adolescent girl would be vaccinated within the next 12 months—the outcome variable. We also examined the relationships between control variables and the outcome. Multivariate logistic regression analysis was conducted to identify variables associated with the caregiver’s perception that the female child was likely to get vaccinated. We use the STATA statistical package for the analysis. Chi-squared tests of independence were conducted at the bivariate level. Adjusted odds ratios from logistic regression analysis were estimated at the multivariate level. We used the Huber–White sandwich estimator to account for clustering of observations in stratified samples by using the STATA cluster command [23].

## 3. Results

Table 1, Column 1, shows frequency distributions of sociodemographic characteristics of the sample. About 41% of caregivers were from the three Northern States, and 59% were from the three Southern States. Most respondents, 61%, were aged 30–39, and 31% were aged 40 and older. About 42% of caregivers in our sample were women, 56% were men, and the remaining 1.6% either did not report their gender or reported their gender as non-binary. The sample was well-educated, with 53% of respondents having a bachelor’s or higher education. Most respondents were from cities (56%) or towns (34%), with only 10% from rural areas. About 27% of the sample were mothers, and 27% were fathers, with aunts (15%) and uncles (22%) comprising a slightly lower proportion of caregivers.

Table 1, Column 2, shows the relationship between sociodemographic variables and the percentage of caregivers who believed that their adolescent girl is very likely to get vaccinated within the next year. Overall, most caregivers (60%) believed that it was very likely that the female child in their care would get vaccinated in the next 12 months. Caregivers from the Northern states were significantly more likely than those from Southern States to believe that their female child would get vaccinated (65% vs. 57%). At the bivariate level, none of the other sociodemographic characteristics were associated with the caregiver’s perception that their female child was very likely to get vaccinated.

Table 2, Column 1, shows frequency distributions of motivation, ability, and other factors associated with vaccine uptake. Column 1 shows that caregivers’ motivation to have their female child vaccinated was high, with 73% of caregivers considering it very important to have their adolescent girl vaccinated. At the same time, caregivers’ level of ability to have their adolescent vaccinated was very low: only 20% of caregivers strongly disagreed with the statement that it was difficult to have their female child vaccinated against HPV. Knowledge of the HPV vaccine was low, with only 18% of caregivers able to correctly respond to four questions asked to gauge their level of knowledge of HPV vaccination. Over a quarter (26%) of respondents had known someone who had cervical cancer, consistent both with the high level of prevalence of cervical cancer and awareness of cervical cancer in Nigeria. Most caregivers (72%) were very likely to discuss HPV vaccination with their family and friends. About one-third of respondents believed that most of their family and friends approved of girls getting the HPV vaccination, an injunctive norm. Nearly one-third of respondents (32%) reported that they had not received a COVID-19 vaccination, a variable that we interpret as a proxy for COVID-19 vaccine hesitancy.

Column 2 of Table 2 shows the relationships between these variables and the outcome. Motivation was associated with the caregiver’s perception that their female child would get vaccinated in the next 12 months: 70% of caregivers who had high motivation believed this compared to 33% of other caregivers. High ability was also associated with a higher proportion of caregivers believing that their female child would get vaccinated in the next 12 months (76%) compared with other caregivers (56%). These findings are consistent with the literature on the FBM [16,18]. There was a significant positive relationship between knowledge of the HPV vaccine and the outcome: 75% of caregivers who had the maximum score of four on the knowledge index believed that their adolescent girl would get vaccinated compared with 50% of caregivers who had a score of 0-1. Caregivers who perceived that their family and friends were supportive of the HPV vaccination were more likely to believe that their female child would get vaccinated (79% versus 52%). Consistent with the idea that COVID-19 vaccine hesitancy may influence HPV vaccine uptake, caregivers who had not received a COVID-19 vaccination were less likely to believe that their adolescent girl was very likely to get vaccinated in the next 12 months (49% vs. 66%).

In Table 3, we present the results of our multivariate logistic regression analysis. This analysis was undertaken to discern which variables were associated with the caregivers’ perception that the female child was very likely to get vaccinated in the next 12 months, after controlling for other variables. The table shows variables that were significant at either the bivariate or the multivariate level were or both. Variables that were not significantly associated with the outcome at both the bivariate and the multivariate levels are not shown in Table 3.

Our analysis shows that motivation and ability retained their significance at the multivariate level. Specifically, caregivers with high motivation had a 2.30 times higher odds ratio of believing that their adolescent girls would receive the HPV vaccine within the next 12 months compared to other caregivers. Similarly, caregivers with high ability were 1.69 times more likely than other caregivers to believe that their female child was very likely to get vaccinated. Additionally, we found that higher levels of knowledge of the HPV vaccine was positively associated with the expectation of the adolescent getting vaccinated.

Table 3 shows that, after adjusting for motivation, ability, and other variables, a caregiver’s knowing someone with cervical cancer was not associated with the expectation that the adolescent girl in their care was very likely to receive an HPV vaccination. However, a caregiver who was very likely to discuss HPV vaccination with family or friends had a 4.78 times higher odds ratio of believing that their adolescent girl would get vaccinated against HPV. The injunctive norm, represented by friends and family members’ approval of HPV vaccination, was associated with a 1.74 times higher odds ratio of a caregiver believing that HPV vaccination was very likely.

Caregivers who had not received a COVID-19 vaccine were less likely to believe that their female child would receive an HPV vaccine within a year, with an adjusted odds ratio of 0.71. After adjusting for other variables in the model, there was no difference by region in the likelihood of a caregiver believing that their child would receive an HPV vaccine. Caregivers in cities were more likely to believe that their female child would get vaccinated in the next 12 months, with a 1.26 times higher odds ratio. Compared to caregivers with primary or secondary education, having a bachelor’s degree was associated with a lower likelihood (aOR = 0.57) of believing that the adolescent would get vaccinated. Overall, the model explained approximately 21% of the variance in the outcome, as indicated by the Pseudo-R2 value”.

The goodness of fit test showed a Pearson chi-squared (1226) value of 1256.09 with a *p*-value of 0.2689. The area under the ROC curve was 0.79, which is considered acceptable.

## 4. Discussion

This study, conducted among caregivers who use Facebook and Instagram in six Nigerian states, provides valuable insights about how motivation, ability, social factors, and knowledge levels of caregivers influence their perception that their female child is likely to get the HPV vaccine. Study findings reveal a complex interplay between these factors, with significant implications for public health interventions aimed at improving HPV vaccine uptake.

The association between motivation and the perceived likelihood of vaccination is consistent with the Fogg Behavior Model (FBM), which posits that, when motivation, ability, and a prompt happen in the same moment, they lead to a behavior. While motivation has a strong association with a caregiver’s expectation of their female child getting vaccinated, it is important to note that caregivers’ motivation to have their female child vaccinated was already quite high at the time of our survey: 72% of caregivers reported that they considered it very important for their female child to get the HPV vaccine.

The finding that a caregiver’s ability was a significant predictor of the likelihood of an adolescent’s getting vaccinated, even after adjusting for other variables, indicates that ability factors may remain a major challenge for HPV vaccine uptake in Nigeria. Particularly since the level of ability of Nigerian caregivers is very low (20%), there is an enormous opportunity for interventions to increase caregivers’ ability. Ability is indeed what the government’s ongoing HPV vaccination campaign is meant to address—making free vaccines available across the country. Efforts to simplify the vaccine adoption process, reduce costs, and increase vaccine access are likely to have a very positive impact on vaccine uptake. While government-sponsored programmatic efforts are underway, it is essential to assess the extent to which the current initiative improves caregivers’ ability to have their children vaccinated. Regular assessments can help inform policy makers and implementers on whether additional, more targeted, interventions are needed to enhance caregivers’ capacity to facilitate vaccine uptake by adolescents.

The strong influence of injunctive norms on caregivers’ beliefs about vaccination suggests that social approval is a potent driver of HPV vaccine uptake among the study participants. Consistent with this finding, the discussion of HPV vaccination with family and friends is associated with a caregiver’s higher perceived likelihood of their female child receiving an HPV vaccination. Interventions that leverage positive family and peer influences and increase discussion of HPV vaccination among members of caregivers’ social networks are likely to be extremely beneficial.

Knowledge of the HPV vaccine was another important predictor of the outcome. This finding aligns with previous research demonstrating the importance of awareness and understanding of what is involved with a behavior in health behavior decisions. Improving knowledge about HPV vaccination through educational campaigns will continue to remain important in increasing HPV vaccine uptake.

While the study findings are broadly consistent with our expectations, the study also suggests an apparent paradox: caregivers with higher education levels were less likely to believe that their female child would be vaccinated within the next year. This counterintuitive result challenges the assumption that higher education correlates with better health practices. It suggests beliefs among more educated Nigerians should be examined more carefully to determine how more educated caregivers may be persuaded to support HPV vaccine adoption. In other words, educated Nigerians who use Facebook and Instagram may form a separate segment whose motivation, ability, and social influences should be considered separately.

Our study presents some of the first empirical evidence that supports the claim that COVID-19 vaccine hesitancy is likely to impact HPV vaccine uptake: caregivers who had not received a COVID-19 vaccination (considered a proxy for COVID-19 vaccine hesitancy) were less likely to expect their female child to get vaccinated for HPV in the next 12 months. The negative association between COVID-19 vaccine hesitancy and HPV vaccine uptake highlights a potential spillover effect of vaccine confidence across different types of vaccinations. Addressing vaccine hesitancy more broadly may, therefore, have a beneficial impact on the acceptance of both HPV and COVID-19 vaccines.

The lack of association between a caregiver’s knowing someone with cervical cancer and their expectation that their female child will get vaccinated in the next 12 months may indicate that knowing someone with the disease does not necessarily translate into preventive action. This finding may reflect a gap between awareness of cervical cancer’s severity and knowledge that HPV vaccination is a preventive measure.

It is interesting that this study did not find any difference in the perceived likelihood of HPV vaccination by region at the multivariate level. At the bivariate level, the findings showed a significantly higher perception of the likelihood that their female child would get vaccinated among caregivers in the north. This disparity, at the bivariate level, was perhaps not surprising given that vaccine-related misinformation has been much higher in Southern Nigeria since the COVID-19 pandemic and the level of trust in the government has been lower. That this disparity became non-significant at the multivariate level after adjusting for motivation, ability, and social factors, suggesting that regional differences in vaccine uptake may be overcome through behavioral interventions. Further exploration to determine whether similar findings emerge from other data is important as it would provide a strong rationale and justification for designing and implementing behavioral interventions.

The odds ratios associated with socio-psychological factors such as motivation, ability, social norms, discussion, and knowledge were larger than those associated with sociodemographic factors such as education, age, or gender. While the use of socio-psychological variables is not new in public health [24], it has not been part of the mainstream approach used by immunization programs in low- and middle-income countries. Our findings suggest that using a behavior model which enables an understanding of the effect of socio-psychological variables on immunization outcomes would greatly increase the effectiveness of immunization interventions in low- and middle-income countries. Finally, the multivariate analysis, which explained a substantial proportion of the variance in caregivers’ expectation of their female child getting vaccinated, suggests that interventions targeting multiple factors may be most effective. For instance, combining educational efforts with strategies to increase social support and reduce logistical barriers could address multiple barriers that may influence HPV vaccine uptake. Future research should explore the underlying reasons for the observed associations and how best to design interventions that address the identified barriers and facilitators of HPV vaccine uptake.

### Strengths and Limitations

Two limitations of our research are noteworthy. The first is the use of a cross-sectional survey. Since predictor variables and the outcome were measured at the same time, no causal inferences can be made from our study. The second is that the study is based on a non-representative sample, which limits its generalizability. A strength of our study is that it is based on a behavior model that has been tested across multiple health behaviors and populations in Nigeria [16,17]. Another strength of the study is that it operationalizes a broad range of variables and adjusts for sociodemographic differences between caregivers before reaching conclusions regarding relationships between variables.

It is worth noting that the recruitment and data collection strategy used in this study is different from that used in the standard representative household survey. Conducting a household survey in Nigeria is a resource- and time-intensive exercise. In our case, this would have meant finding and hiring a survey research organization with expertise in conducting household surveys in Nigeria. We estimate that conducting a household survey in six Nigerian states would have cost us about USD 150,000–200,000, which was far beyond the budget that was available to us. In addition, planning such a survey, listing households, hiring teams to conduct household interviews, and collecting data would have taken several months. By contrast, the digital survey data collection was completed over a few weeks with an advertising cost of a few thousand USD. The main cost incurred in conducting this survey was the cost of our time, which the Behavioral Insights Lab donated for this project.

The great advantage of a representative household survey is that it provides generalizable findings. However, the cost of such an effort is often prohibitive for field-based programs. This usually means that very limited programmatic research is conducted in countries like Nigeria which have limited funds for research. By conducting a considerably less expensive, but non-representative survey, we hoped to provide insights for program designers to develop interventions that may increase the effectiveness of the HPV vaccine rollout in Nigeria.

## 5. Conclusions

Our findings highlight the importance of motivation, ability, and social factors in driving the acceptability of HPV vaccination in Nigeria. It suggests that multi-faceted interventions that influence the ability of caregivers and the social influence upon them are important in addition to efforts to reduce vaccine hesitancy more broadly. Interventions should be designed at multiple levels, with the focus being on helping individuals overcome the range of barriers that they face—many of which exist at a level beyond their individual control.

## Figures and Tables

**Table 1 vaccines-12-00084-t001:** Frequency distributions of caregiver characteristics and cross-tabulations with caregivers’ perception that the child in their care is very likely to be vaccinated in the next 12 months.

	(1) Sample Characteristics (n = 1429)	(2) % of Caregivers Who Believe That Their Female Child Is Very Likely to Have the HPV Vaccination in Next 12 Months (n = 1429)	(3) *p*-Value
Region			
South	59.0%	56.7%	0.001
North	41.0%	65.4%	
Age			
18–29	8.6	59.3%	0.134
30–39	60.5	62.3%	
40+	30.9	56.6%	
Gender of Caregiver			
Man	56.3%	60.5%	0.724
Woman	42.1%	60.2%	
Not reported/non-binary	1.6%	52.2%	
Education			
None, Primary, SSCE, GCE	14.3	63.2%	0.085
OND, HND	32.5	63.4%	
BSc or higher	53.3	57.6%	
Residence			
City	56.0	62.7%	0.092
Town	34.1	57.3%	
Rural	9.9	56.3%	
Relationship to Girl 9–17			
Mother	26.8	61.9%	0.768
Father	26.7	60.9%	
Aunt	14.6	59.1%	
Uncle	21.7	60.3%	
Other	10.3	55.8%	
Total	100%	60.3%	

**Table 2 vaccines-12-00084-t002:** Frequency distributions of caregiver motivation and ability and cross-tabulations with caregivers’ perception that the child is very likely to get vaccinated in the next 12 months.

	(1) Sample Characteristics (n = 1429)	(2) % of Caregivers Who Believe That Their Female Child Is Very Likely to Have the HPV Vaccination in the Next 12 Months (n = 1429)	(3) *p*-Value
Motivation (“Getting the girl who is in my care vaccinated against HPV is important to me”)			
All other responses	27.2%	33.4%	<0.001
Strongly agree	72.8%	70.3%	
Ability (“Getting the girl child vaccinated against HPV is difficult)			
All other responses	79.8%	56.3%	<0.001
Strongly disagree	20.3%	75.9%	
HPV Vaccine Knowledge Score			
0–1	45.1%	49.1%	<0.001
2–3	36.7%	66.7%	
4	18.2%	75.0%	
Have you ever known anyone who had cervical cancer?			
No	73.7%	58.2%	0.008
Yes	26.3%	66.0%	
How likely are you to discuss HPV vaccination with family or friends?			
All other responses	27.9%	26.3%	<0.001
Very likely	72.1%	73.4%	
“Most of my family and friends approve of girls getting vaccinated for HPV”			
All other responses	67.7%	51.5%	<0.001
Strongly agree	32.3%	78.7%	
Have you received a COVID-19 vaccination?			
No	32.3%	48.6%	<0.001
Yes	67.7%	65.8%	
Total	100%	60.3%	

**Table 3 vaccines-12-00084-t003:** Adjusted odds of caregiver’s perception that the child is very likely to get vaccinated within 12 months ^1^.

	Odds Ratio (n = 1429)	*p*-Value
Motivation (“Getting the girl who is in my care vaccinated against HPV is important to me”)		
All other responses	1.00	
Strongly agree	2.30 (1.76–3.00)	<0.001
Ability (“Getting the girl child vaccinated against HPV is difficult)		
All other responses	1.00	
Strongly disagree	1.69 (1.28–2.23)	<0.001
HPV Vaccine Knowledge Score		
0–1	1.00	
2–3	1.55 (1.39–1.72)	<0.001
4	2.08 (1.18–3.66)	0.011
Have you ever known anyone who had cervical cancer?		
No	1.00	
Yes	1.03 (0.75–1.43)	0.841
How likely are you to discuss HPV vaccination with family or friends?		
All other responses	1.00	
Very likely	4.78 (3.80–6.00)	<0.001
“Most of my family and friends approve of girls getting vaccinated for HPV”		
All other responses	1.00	
Strongly agree	1.74 (1.320–2.29)	<0.001
Have you received a COVID-19 vaccination?		
No	0.71 (0.59–0.85)	<0.001
Yes	1.00	
Region		
South	1.00	
North	1.41 (0.93–2.13)	0.104
Residence		
City	1.26 (1.12-1.41)	<0.001
Town, Rural	1.00	
Education		
None, Primary, SSCE, GCE	1.00	
OND, HND	0.78 (0.47–1.32)	0.357
BSc or higher	0.57 (0.35–0.93)	0.023
Pseudo R2	21.24%	

^1^ Age, gender, relationship not shown.

## Data Availability

Data will be made available upon reasonable request to the first author.

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
