# Peer review of "Determinants of Human Papillomavirus Vaccine Acceptance among Caregivers in Nigeria: A Fogg Behavior Model-Based Approach"

_vaccines, 2024, doi:10.3390/vaccines12010084_

Round 1

Reviewer 1 Report

Comments and Suggestions for Authors

The article is interesting; the sample is large enough. 

The introduction and the discussions need to be improved and also references need to be enriched. 

The authors could talk about other kinds of vaccination that are studied in healthcare professionals and in students. 

I could suggest the following papers. 

Rubella Vaccine Uptake among Women of Childbearing Age in Healthcare Settings

Healthcare, Ferrari et al.

Protective Anti-HBs Antibodies and Response to a Booster Dose in Medical Students Vaccinated at Childhood

Vaccines. Coppeta et al.

Author Response

Thank you for your excellent comments. We have taken your comments into account in the attachd file. 

Reviewer 2 Report

Comments and Suggestions for Authors

Abstract: PLease make clear when the survey was conducted. 

Methods of main manuscript: Please clarify when the study was conducted. 

Results: In all 3 tables, I recommend that p-values are shown by the side of each comparison and not at the bottom of the table.  

Author Response

Thank you for your excellent comments. We have taken your comments into account in the attached file.

Reviewer 3 Report

Comments and Suggestions for Authors

Dear authors,

I extend my gratitude for your dedicated work in scientific research. Your article is both captivating and timely. This study thoroughly explores HPV vaccine acceptance among adolescent girls in Nigeria, employing the Fogg Behavior Model. The innovative utilization of social media for recruitment and interviews adds a contemporary edge to the research methodology.

Your findings underscore the importance of tailoring interventions to bolster caregiver motivation and support, offering a promising avenue for enhancing HPV vaccine uptake in Nigeria. The study's broader implications provide valuable insights for crafting targeted public health strategies to tackle the challenges posed by HPV-related cancers in the region.

The substantial number of individuals examined and the robust statistical methods employed enhance the credibility of your research. This article not only stands as a valuable contribution on its own but also lays a foundation for future statistical meta-analysis studies. These studies could either confirm the obtained results or unearth new correlations. Furthermore, the transparent acknowledgment of study limitations adds to the article's scholarly integrity.

Thank you once again for your impactful contribution to the field.

Best regards.

Author Response

We are very grateful for your extremely positive comments on this paper.

Sincerely,

Sohail Agha

(on behalf of co-authors)

Reviewer 4 Report

Comments and Suggestions for Authors

Thank you for the opportunity to read this article. The topic is certainly interesting as it sheds light on contexts that are not typically the focus of this type of research. The subject has the potential to provide valuable insights for implementing and improving vaccination strategies and policies in Nigeria and similar settings. However, the article appears to have some critical issues that, in my opinion, cannot be resolved with a simple round of revision.

The first concern relates to the introduction. The text seems confusing, with paragraphs disconnected from each other, and the citations insufficient to support the individual sections. Regarding the methods, additional issues arise, such as the overall mode of survey distribution. The authors stated, "our goal was to reflect the heterogeneous nature of the country," but by recruiting through Facebook and Instagram, I believe that a significant portion of the population may not be reached (if they do not use social media) or may not be interested in the type of announcement posted on these platforms. This limits reaching everyone and capturing the heterogeneous nature of any country.

Concerning model building, the strategy between univariate and multivariate is not clear. In the latter, it is particularly unclear which variables were studied for inclusion (not those later inserted). It is not clear why adjustments were made only for age and gender and why there were no actual models that study the predictors of the outcomes of interest.

The discussion section suffers from the same issues as the introduction. I am confident that with more careful work and less haste, the authors will be able to produce a paper that has a higher standard for publication dignity.

Comments on the Quality of English Language

The manuscript exhibits several grammatical errors impacting its clarity and precision. Instances of inconsistent verb tense usage are noticeable, creating confusion in the narrative flow. Additionally, there are instances of awkward phrasing and incomplete sentences that impede the reader's comprehension. Some sentences are overly complex, contributing to ambiguity rather than elucidating the intended meaning. In particular, the discussion section contains numerous run-on sentences, hindering the effective communication of ideas.

Author Response

Thank you for your excellent comments. We have taken your comments into account in the attached file.

Sohail Agha

(on behalf of co-authors)

Reviewer 5 Report

Comments and Suggestions for Authors

This manuscript is interesting and should be published after some small changes mentioned below.

1.       Please include the questionnaire text in full, along with response options (and skip patterns, if applicable), as a supplemental annex file.

2.       The advertising cost of $2492.77: what did that include?  Planning?  Programming?  Design?  Administration?  Or were those simply the charges to run the ads?

3.       In section 2.5 re: sample size, please furnish a few more details about the assumed magnitude of change your study will be powered to detect over time.  (“have a sufficient number of respondents from Bauchi and Lagos to compare changes reported by caregivers in these two states to caregivers in the other four states where the HPV vaccine would be introduced later”)  Did the calculation involve formally power to detect an X% change over time in Bauchi and Lagos?  Or was the idea simply to estimate prevalence precisely in this first survey so that a similarly precise estimate from a later survey would hopefully show a significant change?

4.       Did the sampling involve a target or quota of respondents in each of the 120 strata?  If yes, how were those targets set?  Using some census data?  Something else?  And were the targets achieved in every stratum?  Were the stratum targets and sample constructed to yield a self-weighted analysis?

5.       Several questions are listed in lines 127 to 142, but without the response options, it is challenging to follow the operationalization narrative.  Adding the full questionnaire with response options, as suggested above, will help.

6.       Section 2.7: Please list the software you used for the analysis and clarify whether the hypothesis tests and regression analysis accounted for the stratified nature of the sample.  Please clarify precisely what statistical tests were used to assign the *, **, *** in the manuscript tables and clarify whether those tests accounted for stratification or not.  If they did not, they must be re-done with syntax to do so.

7.       Please report the goodness-of-fit of the multivariable logistic regression model, along with the area under the receiver operating characteristic curve.  Two separate summaries of model fit and discrimination.  If the model does not fit the data, then interpreting it is of little value, and the reader will want to know whether it does a good job discriminating between those who are and are not likely to be vaccinated.  Depending on what software you are using, it may be necessary to report goodness of fit on a simpler not-survey adjusted model.  That is acceptable.  Just say so.  But the significance asterisks (*, **, ***) should be based on software that accounts for the stratified sample structure.  (Or if not, the choice to ignore the sample design should be justified in the text.)

8.       The limitations should be a little more clear about how it is non-representative:  draws from persons who use Facebook or Instagram and therefore may miss out on lower income or lower education parents.  If you have data concerning cellphone penetration or FB/Instagram penetration among parents of teenagers, by state, it might be useful to summarize that info.

9.       The first sentence of the conclusion is somewhat overstated.  In the limitations paragraph you just acknowledged that you cannot evaluate causality and in the next paragraph you say that various items are “driving the acceptability”.  You do not know that.  They are “correlated with acceptability”.  And not “in Nigeria” but “among Facebook and Instagram users in six states of Nigeria”.  Please sharpen the language there to be less sweeping.

10.   Minor point: It should be ‘Gavi’ not ‘GAVI’.

11.   Data availability.  De-identified respondent level data and responses should be made available.  Why does the manuscript say that “aggregate” data is available?  Aggregated in what way?  And why?

12.   Optional: The recruitment and data collection strategy here is unusual…has some cost and timeframe advantages and has some disadvantages with regard to representativeness vis-à-vis a more traditional household survey with multi-stage selection.   I suggest you consider devoting a paragraph to factors influencing the decision between fielding this survey in a face-to-face household survey with more rigorously random and representative selection of respondents versus the strategy used here. 

Author Response

(The authors gave the same response as above.)

Round 2

Reviewer 1 Report

Comments and Suggestions for Authors

The paper has been significantly improved after revision 

Reviewer 4 Report

Comments and Suggestions for Authors

Authors fully addressed the my comments.